# Research policies and scientific production: A study of 94 Peruvian universities

**Pablo Alejandro Millones-Gómez**[1,2]*, **Judith Soledad Yangali-Vicente**[1], **Claudia Milagros Arispe-Alburqueque**[1], **Oriana Rivera-Lozada**[3], **Kriss Melody Calla-Vásquez**[1,4], **Roger Damaso Calla-Poma**[2], **Margarita Fe Requena-Mendizábal**[2], **Carlos Alberto Minchón-Medina**[5]

1 Postgraduate School, Universidad Norbert Wiener, Lima, Perú, 2 Faculty of Dentistry, Universidad Nacional Mayor de San Marcos, Lima, Perú, 3 South American Center for Education and Research in Public Health, Universidad Norbert Wiener, Lima, Perú, 4 Universidad San Ignacio de Loyola, Lima, Perú, 5 Faculty of Physical Sciences and Mathematics, Department of Statistics, Universidad Nacional de Trujillo, Trujillo, Perú

* pablodent@hotmail.com

## Abstract

Studies of research policies and scientific production are essential for strengthening educational systems and achieving objectives such as quality improvement. The aim of this study was to evaluate the influence of research policies on the scientific production of public and private Peruvian universities. An observational, descriptive, secondary analysis study of the research policies of 92 universities and two graduate schools licensed by the National Superintendence of Higher Education of Peru (SUNEDU) was conducted for the period from 2016–2020. Scientific publications from educational institutions were collected from Scopus and Web of Science for the study period, and researchers certified by the National Council of Science and Technology of Peru (CONCYTEC) were divided by group and level. Multiple regression analysis was performed using two models. The analysis indicated that research policies did not influence scientific production in Scopus or Web of Science in either 2019 or 2020 (Model I) but that type of management ($p < 0.01$), number of National Scientific, Technological, and Technological Innovation Registry (RENACYT) researchers ($p < 0.001$) and 2016 scientific production ($p < 0.001$) did influence production when these variables were incorporated into the model (Model II). However, time of licensing and management type had no effects. The number of research policies implemented by Peruvian universities and licensed graduate schools was not large. Therefore, it is recommended that project funding policies, research training, and research collaboration be strengthened and that the management capacity of research centers and institutes be increased.

## Introduction

In the 19th century, England was the main country where the first universities emerged, and universities also began to emerge in countries of the European continent and the United States under the humanist model of Von Humboldt. During the 20th century in Latin America,

**Data Availability Statement:** The data underlying the results presented in the study are available from: SUNEDU: https://www.sunedu.gob.pe/

avances-licenciamiento/ Renacyt: http://renacyt. concytec.gob.pe/ Scopus: https://www.scopus. com/ Web of Science: www.webofknowledge.com.

**Funding:** This work was funded by grants from Universidad Norbert Wiener (https://www.uwiener. edu.pe/) (076-2020-R-UPNW (PMG)). The funders had no role in study design, data collection and analysis, decision to publish, or preparation of the manuscript.

**Competing interests:** The authors have declared that no competing interests exist.

science and research were incorporated into universities. In the early 1900s in Peru, reforms began at the National University of San Antonio Abad in Cuzco and at the National University of San Marcos in Lima. These reforms laid the foundations for what are currently considered fundamental functions of academia in university law [1]. Scientific research is an essential function of Peruvian universities according to University Law N° 30220 [2]; in this law, the university is defined as a *"research and teaching-oriented* community," where research can transform and improve teaching processes and generate knowledge [3] and thus contribute to economic development, social development, and the implementation of public policies in the country [4]. This conception represents a paradigmatic change in which science, without losing its rigor, becomes accessible to the public and contributes to a critical, reflective, and upright citizenry that, despite living under varying exogenous factors, composes a well-informed society [5].

"The theory of human capital management supports the need to invest in the training of people to increase productivity"; that is, the process of training highly qualified personnel in scientific research is a substantial challenge for Latin American teachers [6]. Universities not only produce knowledge through research but also have the duty to teach the next generation of professionals and researchers. Strengthening the research capabilities of Peruvian universities is one strategy to increase scientific production. Additionally, universities often struggle to balance academic objectives, such as providing quality training and committing to adopt policies and good research practices that promote a research culture [7].

In their strategies for promoting research, higher education institutions should consider improvements in infrastructure, financial support for research, and the training of researchers to generate quality research [8]. Furthermore, it is important to develop practices in the university environment that allow the dissemination of research results; such dissemination of knowledge will not only provide prestige to institutions because of their scientific production but also generate recognition of universities as leaders and generators of new knowledge and as institutions committed to research and innovation for the development of their countries [9].

Research provides answers to scientific, technological, social, educational, and humanistic questions. Thus, it is possible to develop a multidisciplinary vision that guides the training of ethical and high-quality professionals according to national and international standards. Such an approach can overcome any kind of difficulties since scientific production is an indicator that allows evaluations of the scope of contributions to the development of knowledge [10, 11].

Research policies provide the necessary framework to effectively and efficiently guide the management of research in higher-level educational institutions and study programs [8]. In addition, these policies provide guidelines for budgets, mobility, infrastructure, strategic alliances, and participation in research networks and research centers and institutes [12]. Vice rectors of research are responsible for managing and promoting research policies in Peruvian universities. These administrators are expected to have previous quantifiable research experience evidenced by publications in indexed journals and to have demonstrated their suitability for the position [13]. Currently, as a result of the licensing process conducted by the National Superintendence of Higher Education (Superintendencia Nacional de Educación Superior Universitaria—SUNEDU), universities require certain qualifications of their teachers, such as publications and participation in management, science and technology projects, and scientific events.

Scientific production at the doctoral level in the fields of health sciences and engineering requires researchers' ability to conduct research and generate new knowledge for both their countries of origin and the countries where they studied [14]. In Latin American academic society, a multiplicity of tasks and commitments coexist in higher education institutions that

require academics and scientists to establish collaborations and networks [15]. Research collaboration has contributed to generating international university networks through strategic alliances with prestigious entities [16], and these research networks have generated an increase in scientific production through coauthorship in publications [17].

International collaboration is more efficient than individual work, increasing the potential for publication in high-impact journals through joint research. Universities actively participate as executors and promoters of different research activities; therefore, it is necessary for them to strategically promote these types of policies [18]. Funding policies are also notable because they impact scientific production, to the extent that funding has a positive effect on scientific production [19]. Furthermore, policies related to improving and training human capital in research are of utmost importance and should be implemented at the undergraduate level with so-called research incubators, which should be formalized as spaces that generate debate, analyses, and knowledge formation [20].

For these reasons, scientific production is a relevant topic for universities and an important subject for analysis. There is still a marked difference between the scientific production of "*developing*" and "*developed*" countries; this difference lies in aspects such as the absence of investment, the insufficient allocation of state resources, and the low dissemination of university production [21].

Scientific production in Latin American and Caribbean countries has grown substantially, and Peru is no stranger to the processes of scientific production; however, efforts to achieve scientific development are still insufficient, and Peru is ranked 73rd in the world and eighth in Latin America in scientific production [22].

Current data provided by CONCYTEC [23] indicate that in Peru, only 0.10% of the GDP is invested in research, which is much lower than that in other South American countries, such as Brazil, and lower than the average for Latin America and the Caribbean (1.3 and 0.8, respectively). Likewise, the country contributes only 1% of the scientific publications in the region [24].

It is necessary to establish strategic alliances with the private sector to strengthen research funding policies and increase scientific production in the country [25]. In addition, funding promotes the training of human resources in research and provides monetary support for publications [26]. Additionally, salaries for research teachers, improvements in research infrastructure, and payments to external consultants are associated with an increase in scientific production in health sciences [27].

Previous studies have reported that policies related to research affect the scientific production of and scientific publications by universities [4]. Having adequate research policies provides guidance for best scientific practices, allowing the creation of ecosystems that facilitate the development of researchers and increase the scientific publication culture.

In Latin America, studies have revealed the importance of policies for scientific production. In Peru, a publication modality was adopted as part of the public policies incentivizing scientific research; however, according to recent bibliometric criteria, the levels of scientific production are still incipient [28]. A growing trend in scientific production in Web of Science in Peru and Ecuador was observed due to investment in research and development (R&D) [29]. In licensed universities, scientific production in Scopus was analyzed using multiple linear regression applied to the number of research professors registered in the National Scientific, Technological, and Technological Innovation Registry (Registro Nacional Científico, Tecnológico y de Innovación Tecnológica—RENACYT) and the type of management.

The management of academic and business research includes the identification of lines of research that direct the funds and technical efforts of groups dedicated to research, development, and innovation (R&D&I) activities. In the decision-making process that guides this

management, the opinions derived from the experiences and individual interests of researchers usually prevail over objective consultations with various sources that could provide information to optimize this process. Moreover, lines of research are strategic organizational subsystems employed by the research groups to guide their activities; these subsystems allow the formation of teams of researchers who have similar concerns, interests, and needs in the same field. Lines of research are also considered the guiding axes of research activity. Research teams have theoretical support that allows the integration of knowledge and the continuity of the work of a group of people and institutions committed to the development of knowledge in a specific field [30].

Studies of research policies and scientific production are essential for strengthening educational systems and achieving objectives such as quality improvement. Therefore, this study sought to evaluate the influence of research policies on the scientific production of Peruvian public and private licensed universities using multiple regression models. Although we report findings from Peruvian universities, we hope that our results are relevant to the field of scientific production at the Latin American level.

## Materials and methods

An observational, descriptive, secondary analysis of the research policies of 92 universities and two graduate schools licensed by SUNEDU was conducted for the period from 2016–2020 (https://www.sunedu.gob.pe/lista-de-universidades-licenciadas). The scientific production of educational institutions in Scopus and Web of Science during the study period was examined, and researchers certified by the National Council of Science and Technology (http://renacyt. concytec.gob.pe) were divided by group and level.

The research policies of universities and graduate schools were obtained from each institution's licensing resolution. The selection and categorization of each research policy were performed with consideration of the indicators established in Condition IV of the Regulations of the Licensing Procedure for Public or Private Peruvian Universities of SUNEDU [9]. Policies involving universities providing economic support to their teachers (project funding, special bonuses for research professors, bonuses, and/or publication aid) were considered research funding policies. Human capital policies were those seeking to strengthen the capacities of teachers and students in educational activities (research incubators, recognition of human resources involved in research, mobilization, and training in research). Research collaboration policies were those that linked the university community with other domestic or foreign institutions in the development of research projects (agreements). Policies to promote research centers and institutes were those that sought to group highly trained scientific personnel in lines of research promoted by each institution.

Scientific production indicators were obtained from the Web of Science and Scopus platforms. The search included studies published from 2016–2020, considering the authors' affiliations (original name and variants) with the institutions evaluated as search criteria. The number, group, and level of researchers were searched in RENACYT, which classifies researchers into two groups and different levels according to academic degree, production and relevance of scientific knowledge and/or technology, development of research projects, and training of human resources. The classification consists of two groups: "María Rostworowski," with three levels, and "Carlos Monge Medrano," with four levels.

Scientific production and policies up to August 31, 2020, were considered.

The data analysis included the calculation of the descriptive statistics of the research policies of the universities and licensed graduate schools, the scientific production of the universities and the number of RENACYT researchers at the universities, which are presented in the tables,

as well as the management type (national and private) and time of licensing as of August 30, 2020.

The influence of research policies on the scientific production of public and private Peruvian universities was evaluated using multiple linear regression models. The analysis was performed using EViews, version 10, using the ordinary least squares method and robust estimates of standard errors.

First, the models were evaluated to explain the number of RENACYT researchers in Peruvian universities and graduate schools as an indicator directly related to scientific production; the first model (Model I) included the research policies implemented since 2016, and the second (Model II) also included the type of management, licensing time, and scientific production up to 2016 as explanatory or independent variables. Scientific production up to 2016 was included as an independent variable because it represents the production of the universities prior to the implementation of the policies under study authorized by the university law; it constituted a "baseline" measurement of the dependent variable.

Second, the current scientific production, corresponding to 2019 and 2020 (still to be concluded), was directly evaluated. Scientific production comprised the total publications of the researchers in Scopus and Web of Science. For each year, two regression models were also estimated. The first model (Model I), as in the case of the number of RENACYT researchers, included the research policies implemented since 2016, and the second model (Model II) incorporated, in addition to the type of management, licensing time, and scientific production up to 2016, the number of RENACYT researchers [31].

## Results

The number of financial research policies implemented was not large. A maximum of six research funding policies, eight human capital research policies, two research collaboration policies, and seven policies promoting research centers and institutes were implemented, with each university most frequently only implementing one of these policies (Table 1).

Not all universities and licensed graduate schools had researchers with publications in Rostworowski or Monge; most often, they did not have researchers at the indicated levels. However, there were universities with large numbers of researchers, such as Universidad Nacional Mayor de San Marcos (454 researchers; 277 Rostworowski and 177 Monge), Pontificia Universidad Católica del Perú (299 researchers; 131 Rostworowski and 168 Monge), Universidad Peruana Cayetano Heredia (183 researchers; 103 Rostworowski and 80 Monge), Universidad Agraria la Molina (132 researchers; 88 Rostworowski and 44 Monge), and Universidad Nacional San Agustín (117 researchers, 83 Rostworowski and 34 Monge) (Table 2).

The scientific production of universities and graduate schools, on average, increased between 2016 and 2019, and it is expected that in 2020, production will also have increased. The universities with the greatest scientific production in this period were Universidad Peruana Cayetano Heredia (4057 articles; 2044 in Scopus and 2013 in Web of Science), Pontificia Universidad Católica del Perú (3650 articles; 2096 in Scopus and 1554 in Web of Science), and Universidad Nacional Mayor de San Marcos (3587 articles; 1911 in Scopus and 2044 in Web of Science) (Table 3).

The effect of research policies on the number of RENACYT researchers was examined. Model I included only research policies, whereas Model II included not only research policies but also management type, licensing time (days), and 2016 scientific production.

The multiple regression analysis in Model I indicated that the number of RENACYT researchers was not influenced by research policies. In Model II, which included management

**Table 1. Research policies of Peruvian universities and licensed graduate schools.**

| | | Statistical Indicators | | | | | |
|---|---|---|---|---|---|---|---|
| | | Minimum | Maximum | Mean | Median | Mode | SD |
| Research funding policies | Project funding | 1 | 2 | 1.02 | 1 | 1 | 0.15 |
| | Special bonuses for research professor | 0 | 2 | 1.04 | 1 | 1 | 0.64 |
| | Bonuses and/or publication aid | 0 | 2 | 1.18 | 1 | 1 | 0.66 |
| | Total | 1 | 6 | 3.26 | 3 | 3 | 1.28 |
| Human capital research policies | Research incubators | 0 | 2 | 1.12 | 1 | 1 | 0.55 |
| | Recognition of human resources involved in research | 0 | 2 | 0.98 | 1 | 1 | 0.61 |
| | Mobilization | 0 | 2 | 1.15 | 1 | 1 | 0.64 |
| | Research training | 0 | 2 | 1.14 | 1 | 1 | 0.50 |
| | Total | 0 | 8 | 4.33 | 4 | 4 | 2.04 |
| Research collaboration policies | | 0 | 2 | 1.05 | 1 | 1 | 0.49 |
| Policies that promote research centers and institutes | | 0 | 7 | 1.35 | 1 | 1 | 1.23 |

SD, standard deviation.

Source: SUNEDU, Web of Science and Scopus.

type ($p < 0.05$) and 2016 scientific production ($p < 0.001$), these factors explained the number of researchers; licensing time did not have a significant effect (Table 4).

The effect of research policies on the number of RENACYT researchers was examined. Model I included only research policies, whereas Model II included not only research policies but also management type, licensing time (days), number of RENACYT researchers, and 2016 scientific production. With respect to the type of management, it was estimated that if the universities had implemented the same research policies, had the same licensing time and had the same 2016 scientific production, then the national universities would have had 20.4 RENACYT researchers more than the private universities. In addition, if the universities had implemented the same research policies and had the same licensing time and type of management, that is, with the other variables remaining constant, for each scientific publication as of 2016, there would now be 0.4 RENACYT researchers.

**Table 2. Number of RENACYT researchers from Peruvian universities and licensed graduate schools.**

| | Statistical Indicators | | | | | |
|---|---|---|---|---|---|---|
| | Minimum | Maximum | Mean | Median | Mode | SD |
| Rostworowski | 0 | 277 | 16.87 | 6 | 0 | 35.86 |
| I | 0 | 206 | 12.04 | 4 | 0 | 25.87 |
| II | 0 | 29 | 1.81 | 0 | 0 | 4.36 |
| III | 0 | 46 | 3.02 | 1 | 0 | 6.60 |
| Monge | 0 | 177 | 9.55 | 1.5 | 0 | 26.80 |
| I | 0 | 21 | 0.80 | 0 | 0 | 3.12 |
| II | 0 | 45 | 1.60 | 0 | 0 | 5.76 |
| III | 0 | 95 | 4.48 | 1 | 0 | 13.60 |
| IV | 0 | 35 | 2.68 | 0 | 0 | 5.75 |
| RENACYT (CTI-Vitae) | 0 | 454 | 26.43 | 7 | 0 | 61.33 |

SD, standard deviation.

Source: SUNEDU, Web of Science and Scopus.

**Table 3. Scientific production of Peruvian universities and licensed graduate schools.**

| | Statistical Indicators | | | | | |
|---|---|---|---|---|---|---|
| | **Minimum** | **Maximum** | **Mean** | **Median** | **Mode** | **SD** |
| Scopus | 0 | 2096 | 168.03 | 36.5 | 0 | 384.02 |
| 2016 | 0 | 406 | 23.67 | 3 | 0 | 64.89 |
| 2017 | 0 | 438 | 28.89 | 4 | 0 | 75.28 |
| 2018 | 0 | 476 | 36.00 | 6 | 0 | 84.95 |
| 2019 | 0 | 560 | 48.94 | 12.5 | 0 | 102.58 |
| 2020 | 0 | 304 | 30.53 | 7.5 | 0 | 62.69 |
| Web of Science | 0 | 2013 | 97.72 | 19 | 0 | 311.87 |
| 2016 | 0 | 354 | 15.54 | 2 | 0 | 53.26 |
| 2017 | 0 | 513 | 19.46 | 2 | 0 | 70.29 |
| 2018 | 0 | 422 | 21.32 | 2 | 0 | 70.07 |
| 2019 | 0 | 496 | 26.66 | 6 | 0 | 80.17 |
| 2020 | 0 | 248 | 14.74 | 3 | 0 | 40.75 |

SD, standard deviation.

Source: SUNEDU, Web of Science and Scopus.

**Table 4. Influence of research policies on the number of RENACYT researchers in Peruvian universities and graduate schools.**

| | Number of RENACYT Researchers | |
|---|---|---|
| | **Model I** | **Model II** |
| Constant | 27.32872 | 22.12887*** |
| Research funding policies | 6.35141 | -3.01321 |
| Human capital research policies | -6.24782 | -0.02434 |
| Research collaboration policies | 2.96133 | 1.19908 |
| Policies that promote research centers and institutes | 1.74220 | -0.03207 |
| Type of management[a] | | -20.37640* |
| Licensing time | | 0.00817 |
| 2016 Scientific production | | 0.43857*** |
| $R^2$ | 0.00976 | 0.71123 |
| $R^2$ adjusted | | 0.68773 |
| F | 0.21928 | 30.25933*** |
| Estimation error | 62.38672 | 34.2723 |
| Akaike information criterion | 11.29159 | 9.987817 |
| Durbin-Watson statistic | 2.14903 | 1.935249 |

N = 94. All regressions were performed with EViews 10.

[a]National = 0, private = 1.

*$p < 0.05$,

**$p < 0.01$,

***$p < 0.001$.

Source: EViews 10.

**Table 5. Influence of research policies on the scientific production of the universities and graduate schools in Peru in Scopus and Web of Science.**

| | Number of Articles in Scopus and Web of Science | | | |
| | 2019 | | 2020 | |
| | Model I | Model II | Model I | Model II |
|---|---|---|---|---|
| Constant | 85.09582 | -6.54326 | 53.43613 | 4.32716 |
| Financial research policies | 14.51836 | -0.54093 | 6.33964 | -0.20365 |
| Human capital research policies | -20.76523 | -2.73190 | -10.69629 | -1.39945 |
| Research collaboration policies | 27.23967 | 11.87757 | 15.76258 | 6.05254 |
| Policies that promote research centers and institutes | 3.29996 | 4.19015 | 0.67694 | 1.34213 |
| Type of management[a] | | 9.17855 | | 11.77103 |
| Licensing time | | 0.01216 | | -0.00343 |
| Number of RENACYT researchers | | 0.77216*** | | 0.52969*** |
| 2016 Scientific production | | 1.10272*** | | 0.56297*** |
| $R^2$ | 0.01241 | 0.95689 | 0.01167 | 0.91567 |
| $R^2$ adjusted | -0.03198 | 0.95283 | -0.03275 | 0.90773 |
| F | 0.27951 | 235.84490*** | 0.26280 | 115.36080*** |
| Estimation error | 178.38760 | 38.13673 | 100.25380 | 29.96676 |
| Akaike information criterion | 13.25752 | 10.21108 | 12.10501 | 9.72890 |
| Durbin-Watson statistic | 1.94132 | 2.22561 | 1.87048 | 1.86532 |

N = 94. All regressions were performed with EViews 10.

[a]National = 0, private = 1.

*$p < 0.05$,

**$p < 0.01$,

***$p < 0.001$.

Source: EViews 10.

Eliminating the terms that did not reach statistical significance in Model II yielded the estimated reduced model:

$$irenacyt = 17.01436 - 15.85296 \ tg + 0.44644 \ sw16$$

$$(5.01295) *** \ \ (6.98004) * \ \ (0.0320) ***$$

where *irenacyt* is the number of RENACYT researchers, *tg* is management type and *sw16* is the scientific production in Scopus and Web of Science in 2016. The standard errors obtained are in parentheses, and $R^2$ = .70635, $R^2$ adjusted = 0.699892 and AIC = 9.898209. The robustness in the presence of heteroscedasticity remained constant, and the *sw16* coefficient retained the same significance levels using the White, White-Hinkley and MacKinnon-White methods, but the *tg* coefficient remained constant only with the White method.

Furthermore, the multiple regression analysis in Model I indicated that research policies did not influence scientific production in Scopus and Web of Science in either 2019 or 2020. In Model II, which incorporated number of RENACYT researchers ($p < 0.001$) and 2016 scientific production ($p < 0.001$), these factors explained scientific production on the examined databases; licensing time and management type were not significantly correlated with scientific production (Table 5). Similar to the previous models, if the rest of the variables were to remain constant, it was estimated that for each RENACYT researcher, there would be 0.87 publications in 2019 and 0.52 publications in 2020 (up to August 31, 2020), and that for each RENACYT publication in 2016, there would be 1.1 publications in 2019 and 0.56 in 2020.

The same as for the number of RENACYT researchers, for scientific production in Scopus and Web of Science (sw), the elimination of the terms that did not reach statistical significance in Model II led to the estimated reduced models:

$$sw19 = 16.30471 \ tg + 0.852510 \ irenacyt + 1.09721 \ sw16$$

$$(5.73021) ** \ (0.11106) *** \ (0.06132) ***$$

$$sw20 = 14.69354 \ tg + 0.55275 \ irenacyt + 0.55247 \ sw16$$

$$(4.43190) ** \ (0.08590) *** \ (0.04743) ***$$

In both periods, 2019 and 2020, the elimination of the constant allowed the incorporation of the type of management into the estimated multiple linear regression models, with coefficients of determination ($R^2$ values) of 0.95470 and 0.91414 and AIC values of 10.13297 and 9.61912, respectively. The levels of significance were maintained, with robust estimates of standard errors calculated by the White, White-Hinkley and MacKinnon-White methods.

## Discussion

Scientific research is a fundamental pillar for human progression. Therefore, dedication from authorities is required for the generation of new knowledge that can be applied for the benefit of society. However, this strategic orientation may not be consistent with the research policies of universities and licensed graduate schools in Peru. Regarding financial policies for research, the study found that six policies have been implemented, including the funding of projects, special bonuses for research professors, and bonuses and/or publication aid. This result confirms the finding by Aguado-López and Becerril-García [19] that there is an insufficient allocation of state resources to research, which affects the promotion of research, with subsequent effects on infrastructure, financial support, and training [7, 17]. This result calls for synergistic action by the state, companies, and society to implement a comprehensive sustainable development policy. In particular, in the field of business, which corresponds to the private sector, the university should play a more active role; that is, as an important aspect of social responsibility, research should be supported.

Additionally, research policy related to human capital, such as research incubators, recognition of human resources involved in research, mobilization, and research training, with an average implementation of four policies, constituted the most frequent policy type. This result indicates, consistent with the perspective of Numa-Sanjuan and Delgado [18], that human capital research policies should focus on research incubators. Along the same lines, Barros-Bastidas and Turpo-Gebera [6] agreed that the prioritization of investments in training human resources increases productivity. While this is true, policies of this type are long term. However, the position of the National Superintendence of Higher Education (SUNEDU) toward universities is broader, linking strategic and tactical aspects through institutional licensing (2015) and facilitating formative research with basic and applied research developed by research centers; that is, the strategy establishes links between research conducted in undergraduate and graduate programs, a line of continuity that guarantees new pools of researchers in this process. The implementation of this policy requires a financial policy to ensure the sustainability of research projects over time, a decision that corresponds to the coordinated direction of CONCYTEC [23], public and private universities, and business organizations to promote mobility based on the strategic guidelines of territorial planning established by CEPLAN, which are fundamentally oriented to diversify the lines of research. From this

perspective, as argued by García et al. [15], it is important to promote mobility in the doctoral and postdoctoral fields because they represent important strategic aspects to generate new knowledge and improve scientific production, that is, investing in research training for first-line personnel, as Barros-Bastidas and Turpo-Gebera [6] pointed out. In this sense, mobilization and training constitute recognition for the contributions made by outstanding researchers because, in this way, human capital is qualified through the assimilation of new knowledge.

Among the RENACYT researchers from universities and graduate schools with publications in Rostworowski and Monge by CONCYTEC [19], 703 researchers were affiliated with state universities (UNMSM, Universidad Agraria, and Universidad San Agustín), and 482 researchers were affiliated with private universities (PUCP and Cayetano Heredia). That is, of a total of 1185 RENACYT researchers, 59.32% were affiliated with national universities, and 40.68% were affiliated with private universities. However, in the national spectrum, not all universities and graduate schools had researchers with publications in Rostworowski and Monge. As argued by Aguado-López and Becerril-García [19], in developing countries, the limited allocation of funds gives rise to little dissemination of scientific production, preventing the appropriate development of lines of research. Hence, the efforts made by universities regarding scientific production are insufficient. These results reflect the reality of Peru and Latin America and the Caribbean in terms of scientific production, as found in a study by CONCYTEC [23], indicating that in Peru, 0.10% of the GDP is invested in research. Bommann and Mutz [32] noted that Peru contributes only 1.0% in terms of scientific production, and Scimago [21] found that Peru ranks 73rd in the world and eighth in Latin America in scientific production. This harsh reality is an indicator of the focus of official research in Peru; however, this does not prevent recognition of the notable efforts of public and private universities regarding research. Reversing this situation requires establishing strategic alliances with the private sector, i.e., the promotion of applied research with companies; at the international level, collaboration networks that generate an increase in scientific production must be established.

Regarding the scientific production of Peruvian universities and licensed graduate schools, the variation rate in publications in Scopus for the period from 2016–2019 was 37.9%, i.e., an increase in the average number of publications from 24 in 2016 to 49 in 2019. In 2020, due to the effect of the pandemic, there has been a decrease. Regarding publications in Web of Science, the variation rate was 40.1% for the period from 2016–2019; there was an average of 16 publications in 2016 and 27 publications in 2019. Therefore, of the total of 4109 publications retrieved from 2016 to 2020, 51.0% were obtained from the Scopus database, and 49.0% were obtained from Web of Science. Additionally, the universities with the highest volume of scientific production were Cayetano Heredia University, followed by PUCP and UNMSM [21].

Consequently, scientific research should be promoted; as Londoño et al. [14] argued, such promotion implies fundamentally supporting collaborative relationships among various research networks at the international level, that is, promoting mutual aid, which leads to institutions, universities, and research centers prioritizing their international collaboration policies, thus stimulating better management of international relations offices of public and private universities and boosting the potential for publication in indexed journals and databases. A line of action promoted by Sebastián [16] focuses on political actions aimed at improving the quality of publications so that they are both relevant and pertinent. In this regard, the proposal by Lee [9], which recommends concrete actions to promote the dissemination of results, is a key aspect for universities to become leaders in assimilating good practices in the generation of knowledge. In addition, Bollini and Sarthou [26] suggested the development of strategic alliances by universities to encourage their participation in disciplinary, interdisciplinary, and transdisciplinary research [33–36] networks to publicize their results.

Regarding the influence of research policies on the number of RENACYT researchers in Peruvian universities and licensed graduate schools, in Model I, composed of the variables financial research policies, human capital research policies, research collaboration policies, and policies that promote research centers and institutes, the various policies did not influence the number of RENACYT researchers, showing an $R^2$ of 0.976%; that is, the policies did not explain the presence of researchers. This result may have occurred because policies are norms or rules established by an institution that are extended to others so that policies are fulfilled; that is, research-oriented institutions are responsible for executing policies, which involved managing the research, which is not performed uniformly in the various institutions. Therefore, the explanatory capacity of the variables was null. When the variables management type, license time, and 2016 scientific production were introduced into Model II, management type, time of licensing (days), and 2016 scientific production exerted some influence, as was the case for management type ($p < 0.05$) and 2016 scientific production ($p < 0.001$); on the other hand, time of licensing was not significant. Therefore, the explanatory capacity of the management type and 2016 scientific production variables was high because these variables explained 71.12% of the variation in the number of researchers. Eliminating the variables with effects that did not reach statistical significance led to a slightly lower explanatory power of the model of 70.6% but a fit index 70.6% higher than the previous index of 68.88% and a slightly improved AIC value of 9.979; in addition, the estimates were robust in the presence of heteroscedasticity according to the method by White for all coefficients.

This result allows us to conclude that an organization's management approach is a good reference for predicting the development of the investigative skills of researchers who teach, the development of research, and the development of scientific production. This finding is consistent with the fact that a teacher who publishes in a Web of Science or Scopus journal is influenced by factors that have direct application to the design of public policies [37]. Likewise, to the extent that databases update scientific article production, they disseminate results that contribute to systematic reviews and, therefore, raise the quality of scientific production, which places a focus on the adequate management of human talent, with the starting point being undergraduate students. Finally, some universities have managed to obtain licenses because they have met the basic quality conditions established by SUNEDU; however, in relation to condition IV, which refers to lines of research to be developed, teachers who perform research and produce documents and research projects have been directly linked to SUNEDU and CONCYTEC [23], which is why a study focusing on the records of research projects declared by universities is needed. Therefore, as established by Numa-Sanjuan and Delgado [18], it is important to implement policies that involve training human talent in research incubators.

The influence of research policies on the number of articles affiliated with Peruvian universities and licensed graduate schools in Scopus and Web of Science was investigated using two models. Model I included financial policies, human capital research policies, research collaboration policies, and policies that promote research centers and institutes. Model II, in addition to containing the defined variables, included management type, time of licensing, number of RENACYT researchers, and scientific production. Thus, Model I revealed, for both 2019 and 2020, the absence of an influence on the number of articles affiliated with Peruvian universities and licensed graduate schools in Scopus and Web of Science. This result allows us to infer that the declaration of public research policies is not sufficient and that action is necessary. That is, activities must be oriented to a strategic objective; therefore, research management is important.

However, when the additional variables management type, licensing time, number of RENACYT researchers, and 2016 scientific production were introduced, for Model II-2019 and Model II-2020, significant p values were found for the number of RENACYT researchers

(p < 0.001) and 2016 scientific production. Thus, for Model II for the year 2019, the explanatory capacity of the variables was high, as they explained 95.68% of the variability in scientific production; with the same model for the year 2020, the explanatory capacity of the variables was high, as they explained 91.56% of the variability in scientific production. The explanatory capacity improved even more when the model constant was eliminated, allowing the type of management (public or private) to be included in the models and yielding a difference of just over 14 articles, favoring private institutions. An improvement in the goodness of fit was found, with lower Akaike information criteria (a model is better when this coefficient is lower), and the robustness of the model was verified in conditions of heteroscedasticity, where the values of the robust standard errors increased (as usually they do). There was no emphasis on the coefficients of determination because they may not be appropriate in regression models based on the origin.

Therefore, it can be inferred that the number of RENACYT researchers and 2016 scientific production significantly explained the number of articles in Scopus and Web of Science. The influence of the two variables (number of RENACYT researchers and 2016 scientific production) is notable because the variables represent elements of human talent management developed by both public and private universities to promote scientific research and the production of new knowledge, which is later disseminated through Scopus and Web of Science. In relation to the influence of the number of RENACYT researchers, the results agree with the findings by Araujo [31], who also found an influence of the type of management, which in his study favored private management. On the other hand, it was found that the licensing time of universities did not explain scientific production and apparently was contradictory to scientific production in Web of Science, which showed a positive annual increase over time, as shown by Limaymanta et al. [29] for Ecuador and Peru (2009–2018). The inclusion of the variable 2016 scientific production probably subsumed its effect.

As Moquillaza Alcántara [21] pointed out, because scientific production is driven by human capital, salaries for research teachers and the stimulation of increased scientific production are imperative; it is necessary to create appropriate conditions, such as funding and infrastructure. This idea was echoed by Barros-Bastidas and Turpo-Gebera [6], who indicated the importance of investing in people because only in this way can the quality of scientific research and production improve [38–42].

However, the study sought to provide an objective evaluation based on the information obtained from the databases. Notably, one of the limitations of this study is that the development and application of policies in universities depend on the authorities, budget, and context of the moment. Apparently, the need to achieve licensing in each institution forced all universities to implement policies that have generally improved their research indicators.

The institutional licensing system of Peruvian universities aims to ensure appropriate indicators of scientific production. Consequently, in recent years, there has been growth in these indicators. However, these policies have been focused only on basic and general aspects, ignoring the peculiarities of the environment of each university and the contributions they should make to understand and/or work with the system. In this sense, it is recommended that the lines of research align with national policies, consider the priorities of each region of the country, and reduce the gap between university research and needs of the environment.

## Conclusions

The number of research policies implemented by Peruvian universities and authorized graduate schools is not large. Therefore, it is recommended that universities and graduate schools

fund projects, increase research training, and promote research collaboration policies to increase the management capacity of research centers and institutes.

Not all universities and licensed graduate schools had researchers with publications in Rostworowski or Monge; however, there was a group of public and private universities with high numbers of such researchers. Therefore, it is recommended that universities and graduate schools expand research coverage through more active participation in joint projects with other universities and research centers through the creation of strategic alliances and the assimilation of research networks, supporting the use of the explanatory model to study the trends in the evolution of the number of articles in scientific databases.

## Author Contributions

**Conceptualization:** Pablo Alejandro Millones-Gómez, Kriss Melody Calla-Vásquez.

**Data curation:** Pablo Alejandro Millones-Gómez, Judith Soledad Yangali-Vicente, Claudia Milagros Arispe-Alburqueque, Kriss Melody Calla-Vásquez.

**Formal analysis:** Pablo Alejandro Millones-Gómez, Oriana Rivera-Lozada, Carlos Alberto Minchón-Medina.

**Funding acquisition:** Pablo Alejandro Millones-Gómez.

**Investigation:** Pablo Alejandro Millones-Gómez, Claudia Milagros Arispe-Alburqueque, Kriss Melody Calla-Vásquez.

**Methodology:** Pablo Alejandro Millones-Gómez, Judith Soledad Yangali-Vicente, Claudia Milagros Arispe-Alburqueque, Oriana Rivera-Lozada, Kriss Melody Calla-Vásquez, Carlos Alberto Minchón-Medina.

**Project administration:** Pablo Alejandro Millones-Gómez, Oriana Rivera-Lozada.

**Resources:** Pablo Alejandro Millones-Gómez.

**Software:** Pablo Alejandro Millones-Gómez, Carlos Alberto Minchón-Medina.

**Supervision:** Pablo Alejandro Millones-Gómez.

**Validation:** Pablo Alejandro Millones-Gómez, Claudia Milagros Arispe-Alburqueque, Roger Damaso Calla-Poma, Margarita Fe Requena-Mendizábal, Carlos Alberto Minchón-Medina.

**Visualization:** Pablo Alejandro Millones-Gómez, Oriana Rivera-Lozada, Kriss Melody Calla-Vásquez, Roger Damaso Calla-Poma, Margarita Fe Requena-Mendizábal.

**Writing – original draft:** Pablo Alejandro Millones-Gómez, Judith Soledad Yangali-Vicente, Roger Damaso Calla-Poma, Margarita Fe Requena-Mendizábal.

**Writing – review & editing:** Pablo Alejandro Millones-Gómez, Roger Damaso Calla-Poma, Margarita Fe Requena-Mendizábal.

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
