## [Decision Letter · Decision Letter 0]

3 Mar 2021

PONE-D-20-37631

Research policies and scientific production: A study of 94 Peruvian universities

PLOS ONE

Dear Dr. Millones-Gómez,

Thank you for submitting your manuscript to PLOS ONE. After careful consideration, we feel that it has merit but does not fully meet PLOS ONE’s publication criteria as it currently stands. Therefore, we invite you to submit a revised version of the manuscript that addresses the points raised during the review process.

We look forward to receiving your revised manuscript.

Kind regards,

Isabel Novo-Cortí

Academic Editor

PLOS ONE

Additional Editor Comments:

This paper's topic seems to be interesting, but it needs a deep revision, particularly for the theoretical background. Besides, the authors must properly justify the method and its convenience and fitness in this context. The comments of both reviewers point to this weakness of this paper. A major revision is necessary to achieve the academic standards of PLOS ONE.

I strongly recommend to the authors reading the reviewers' recommendations carefully and following their advice and suggestions.

Journal Requirements:

Reviewers' comments:

Reviewer's Responses to Questions

**Comments to the Author**

1. Is the manuscript technically sound, and do the data support the conclusions?

Reviewer #1: Partly

Reviewer #2: Partly

2. Has the statistical analysis been performed appropriately and rigorously? 

Reviewer #1: I Don't Know

Reviewer #2: Yes

3. Have the authors made all data underlying the findings in their manuscript fully available?

Reviewer #1: Yes

Reviewer #2: Yes

4. Is the manuscript presented in an intelligible fashion and written in standard English?

Reviewer #1: Yes

Reviewer #2: Yes

5. Review Comments to the Author

Reviewer #1: The authors’ analysis of the influence of research policies on the scientific production of Peruvian universities was interesting, but several important issues need revision.

1. Firstly, an improvement of this article is adding a section about the literature review of the relevant publications (books, scholarly articles and any other sources) relevant to this topic. In this case, as a recommendation, you may have a look at the article: Acosta Roa, E. R., Marin Velasquez, T. D., & Gonzales Caycho, A. M. (2020). Policies for scientific production in Latin America: Peru, a case study. Revista Ciencias Pedagogicas E Innovacion, 8(1), 62–69. https://doi.org/10.26423/rcpi.v8i1.350.

2. The choice of using variables from a specific year needs to be explained (for instance, why did the authors choose the scientific production from 2016 as an independent variable instead of using the scientific production from 2018?).

3. Please provide a justification for using this methodology and link it to other studies that used the same method.

4. Although the section with the interpretation of the results is lengthy, some coefficients should be explained better (for example, in Table 4 what does it mean the effect of the variable entitled Type of management on the number of RENACYT researchers?).

5. There are several errors in the text (on page 6, the link with the source of the research policies is not working). Also, please check the punctuation (for example, on page 7 in the paragraph about the classification of research policies) and pay attention to the repetitions (the last paragraph from page 12 is redundant since it has similar ideas presented in the last paragraph from page 11).

6. The section with the conclusions seems to be a reiteration of the sections about results and discussion plus several recommendations. An improvement will be to state in a more convincing way the significance of your contribution to scientific research.

Reviewer #2: RESEARCH POLICIES AND SCIENTIFIC PRODUCTION: A STUDY OF 94 PERUVIAN UNIVERSITIES

Authorss: P. A. MILLONES G. (U. Wiener y UNMSM), J. Yangali (U. Wiener), C. Arispe (U. Wiener), O. Rivera (U. Wiener), K. Calla (U. Wiener), M. Requena (UNMSM) y C. Minchón (U. Nacional de Trujillo).

1 ABOUT THE AUTHORS:

- I am surprised about the number of them: 8.

- Most of them are top professionals. Still they are mostly from medicine areas and teaching in universities. It may be an advantage to learn about the institutions. Although it may bias their view about possibilities of academic research in other non-medicine disciplines.

2 PROVIDE THE EDITORS WITH AN EXPERT OPINION REGARDING:

- THE VALIDITY OF THE MANUSCRIPT:

I consider this is an interesting topic and the paper is pertinent.

- THE QUALITY OF THE MANUSCRIPT:

This is a good paper, although appears to be basic. Being eight authors, they may include a wide discussion of the importance of the research for countries like Peru.

I would like to see more discussion about te importance of the topic at least in Latin America

I think it is key for this paper to say something about the comparison of universities with diverse offer of careers and research possibilities.

- SUPPLY AUTHORS WITH EXPLICIT FEEDBACK ON HOW TO IMPROVE THEIR PAPERS, SO THAT THEY WILL BE ACCEPTABLE FOR PUBLICATION IN PLOS ONE. �

The authors might improve the theoretical and/or empirical importance of the topic under study.

The long list of references is not used for a good discussion in the initial sections. What is the state of the art in this topic? Why is key for the development of a country to count on good research production, moreover in Peru?

Also, some arguments must be included about the pertinence o using linear regression models to test such implicit hypothesis in this study.

Additional comments and suggestions are included at the end of this report.

3 KEY ISSUES TO ANSWER:

• What are the main claims of the paper and how significant are they for the discipline?

As much as I understand, this paper claims “research policies influence on the scientific production of Universities”. This is a pertinent but I think too general claim.

• Are the claims properly placed in the context of the previous literature? Have the authors treated the literature fairly?

- It would be useful to discuss some literature, ietiher in theoretical or moreover in empirical terms. References about how similar studies are carried on for other countries would help to better understand the model proposed. The Latin American experience might help, mainly the experience of countries similar to Peru

- Some literature review may help to discuss who make the decisions about the research policies and how those selected indicators of funding and human capital represent those decisions

- Also some literature review would be pertinent to understand how the differences in the career offers may influence in the scientific outcomes.

• Do the data and analyses fully support the claims? If not, what other evidence is required?

It is not clear for me the data used in the analysis. I know the Sunedu y Concytec provide detailed data for universities about some of the variables. Still I assume the authors may have completed their database with institutional information from the universities studied.

According to the methodology applied, I think the results are consistent and significant. Provided that the endogeneous and exogeneous data are mostly quantitative.

• PLOS ONE encourages authors to publish detailed protocols and algorithms as supporting information online. Do any particular methods used in the manuscript warrant such treatment? If a protocol is already provided, for example for a randomized controlled trial, are there any important deviations from it? If so, have the authors explained adequately why the deviations occurred?

I would ask about the main sources of the detailed data used about the human capital and research funding policies by each university included in the analysis (Table 1).

• If the paper is considered unsuitable for publication in its present form, does the study itself show sufficient potential that the authors should be encouraged to resubmit a revised version?

I consider that if previous observations are answered, the paper might be published. There are few efforts and it might contribute to continue studies like this, for enriching the research policies in our countries.

• Are original data deposited in appropriate repositories and accession/version numbers provided for genes, proteins, mutants, diseases, etc.?

Not applicable for this case

• Are details of the methodology sufficient to allow the experiments to be reproduced?

As indicated previously, data availability must be discussed explicitly.

• Is the manuscript well organized and written clearly enough to be accessible to non-specialists?

Yes, in general terms. Still, some edition observations and suggestions:

o The first sections might quote explicitly authors and citations, to know how far the proposals are based on previous contributions.

o Tables must have some reference about the specific source of data used, and it is important in each table.

o More discussion after the results, referring how far the results of each table helps to improve the research outcomes in Peru

o The final discussion is more referred to the policy institutions and authorities which is fine. Still some of these results have also being influenced by the own decisions of individual universities and authorities. It would be helpful to indicate such observations.

o The References section is poorly presented. At least, the section might present authors under alphabetical order of their last names.

4 FINAL COMMENT

Although confidential comments to the editors are respected, any remarks that might help to strengthen the paper should be directed to the authors themselves.

From my view, because it is a muldisciplinary study about research in higher education institutions, it is important that the study be published after those previous observations.

PROF. JANINA LEÓN CASTILLO

Departamento de Economía – Jefa

Pontificia Universidad Católica del Perú

6. PLOS authors have the option to publish the peer review history of their article (what does this mean?). If published, this will include your full peer review and any attached files.

Reviewer #1: No

Reviewer #2: **Yes: **Janina V. Leon Castillo

---

## [Author Response · Author response to Decision Letter 0]

24 Mar 2021

Dear Editor and reviewers, I am returning the manuscript with the requested corrections.

---

## [Decision Letter · Decision Letter 1]

27 Apr 2021

PONE-D-20-37631R1

Research policies and scientific production: A study of 94 Peruvian universities

PLOS ONE

Dear Dr. Millones-Gómez,

Thank you for submitting your manuscript to PLOS ONE. After careful consideration, we feel that it has merit but does not fully meet PLOS ONE’s publication criteria as it currently stands. Therefore, we invite you to submit a revised version of the manuscript that addresses the points raised during the review process.

We look forward to receiving your revised manuscript.

Kind regards,

Isabel Novo-Cortí

Academic Editor

PLOS ONE

Journal Requirements:

Additional Editor Comments (if provided):

The new version of the paper has clearly been improved. Nevertheless, the authors should fix some minor issues, following the reviewers' suggestions.

Reviewers' comments:

Reviewer's Responses to Questions

**Comments to the Author**

1. If the authors have adequately addressed your comments raised in a previous round of review and you feel that this manuscript is now acceptable for publication, you may indicate that here to bypass the “Comments to the Author” section, enter your conflict of interest statement in the “Confidential to Editor” section, and submit your "Accept" recommendation.

Reviewer #1: All comments have been addressed

Reviewer #2: All comments have been addressed

2. Is the manuscript technically sound, and do the data support the conclusions?

Reviewer #1: Yes

Reviewer #2: Partly

3. Has the statistical analysis been performed appropriately and rigorously? 

Reviewer #1: Yes

Reviewer #2: Yes

4. Have the authors made all data underlying the findings in their manuscript fully available?

Reviewer #1: Yes

Reviewer #2: Yes

5. Is the manuscript presented in an intelligible fashion and written in standard English?

Reviewer #1: Yes

Reviewer #2: Yes

6. Review Comments to the Author

Reviewer #1: The authors submitted an improved version of their article. Although I expected to see an additional section about the literature review, I suppose that the relevant publications on this subject have already been mentioned in the introduction. Besides, to control for heteroskedasticity, the authors should report if the standard error of the regressions are robust. Also, some spelling errors need to be corrected (for example p.10 “firstly”, “secondly”). Overall, the new explanations included in this version added more clarity to the article.

Reviewer #2: Beyond issues of style, I consider there are room for still more improvement in the research paper. The new versión has incorporated several of my observations. Some issues I still would like to see in the paper are about the data analysis, to be positive that the model applied here is the most adequated. Also some policy recommendatios may be useful at the end of the document. And some edition suggestions (e.g., in the Reference section) I made before, may help in the final version.

7. PLOS authors have the option to publish the peer review history of their article (what does this mean?). If published, this will include your full peer review and any attached files.

Reviewer #1: No

Reviewer #2: No

---

## [Author Response · Author response to Decision Letter 1]

12 May 2021

Dear reviewers, I am returning the manuscript with the requested corrections.

---

## [Editor Report · Decision Letter 2]

17 May 2021

Research policies and scientific production: A study of 94 Peruvian universities

PONE-D-20-37631R2

Dear Dr. Millones-Gómez,

We’re pleased to inform you that your manuscript has been judged scientifically suitable for publication and will be formally accepted for publication once it meets all outstanding technical requirements.

Kind regards,

Isabel Novo-Cortí

Academic Editor

PLOS ONE
---

## [Editor Report · Acceptance letter]

20 May 2021

PONE-D-20-37631R2 

Research policies and scientific production: A study of 94 Peruvian universities 

Dear Dr. Millones-Gómez:

I'm pleased to inform you that your manuscript has been deemed suitable for publication in PLOS ONE. Congratulations! Your manuscript is now with our production department. 

Kind regards, 

on behalf of

Dr. Isabel Novo-Cortí 

Academic Editor

PLOS ONE